# Novel Derivatives of Eugenol as a New Class of PPARγ Agonists in Treating Inflammation: Design, Synthesis, SAR Analysis and In Vitro Anti-Inflammatory Activity

**DOI:** 10.3390/molecules28093899

**Published:** 2023-05-05

**Authors:** Noor Fathima Anjum, Dhivya Shanmugarajan, B. R. Prashantha Kumar, Syed Faizan, Priya Durai, Ruby Mariam Raju, Saleem Javid, Madhusudan N. Purohit

**Affiliations:** 1Department of Pharmaceutical Chemistry, Farooqia College of Pharmacy, Mysuru 570 015, India; nfanjum79@gmail.com (N.F.A.); stargm012@gmail.com (S.J.); 2Department of Pharmaceutical Chemistry, JSS College of Pharmacy, JSS Academy of Higher Education & Research, Mysuru 570 015, India; dskeerthanasai@gmail.com (D.S.); brprashanthkumar@jssuni.edu.in (B.R.P.K.); sfaizan313@gmail.com (S.F.); priyampharm3@gmail.com (P.D.); rubyrajan7@gmail.com (R.M.R.)

**Keywords:** eugenol derivatives, molecular docking, pharmacophore, structure–activity relationship, anti-inflammatory activity

## Abstract

The main objective of this research was to develop novel compounds from readily accessed natural products especially eugenol with potential biological activity. Eugenol, the principal chemical constituent of clove (*Eugenia caryophyllata*) from the family Myrtaceae is renowned for its pharmacological properties, which include analgesic, antidiabetic, antioxidant, anticancer, and anti-inflammatory effects. According to reports, PPARγ regulates inflammatory reactions. The synthesized compounds were structurally analyzed using FT-IR, ^1^HNMR, ^13^CNMR, and mass spectroscopy techniques. Molecular docking was performed to analyze binding free energy and important amino acids involved in the interaction between synthesized derivatives and the target protein. The development of the structure–activity relationship is based on computational studies. Additionally, the stability of the best-docked protein–ligand complexes was assessed using molecular dynamic modeling. The in-vitro PPARγ competitive binding Lanthascreen TR-FRET assay was used to confirm the affinity of compounds to the target protein. All the synthesized derivatives were evaluated for an in vitro anti-inflammatory activity using an albumin denaturation assay and HRBC membrane stabilization at varying concentrations from 6.25 to 400 µM. In this background, with the aid of computational research, we were able to design six novel derivatives of eugenol synthesized, analyzed, and utilized TR-FRET competitive binding assay to screen them for their ability to bind PPARγ. Anti-inflammatory activity evaluation through in vitro albumin denaturation and HRBC method revealed that **1f** exhibits maximum inhibition of heat-induced albumin denaturation at 50% and 85% protection against HRBC lysis at 200 and 400 µM, respectively. Overall, we found novel derivatives of eugenol that could potentially reduce inflammation by PPARγ agonism.

## 1. Introduction

Nuclear hormone receptor is also referred to as peroxisome proliferator-activated receptors (PPARs). PPARα, PPARβ (also called PPARδ), and PPARγ are three different subtypes [1]. They are crucial for the metabolism of glucose and lipids, as well as for the proliferation and differentiation of cells. The liver, kidney, and intestinal mucosa possess high levels of PPARα which is involved in the catabolism of fatty acids. PPARβ is ubiquitously expressed and it plays a role in cell differentiation and metabolism of lipids [2]. The target protein PPARγ is said to regulate inflammatory responses in the physiological systems of the body. The PPARγ has been shown to be essential for controlling inflammatory and immunological responses. PPARγ is abundantly expressed in dendritic cells, macrophages, adipose tissue, and microglia. Numerous inflammatory factors, including Tumor Necrosis factor alpha (TNF-α), interleukin-6 (IL-6), interleukin-8 (IL-8), cyclooxygenase (COX2), and inducible nitric oxide synthase, can be activated by nuclear factor-kappa B (NF-κB). Reactive oxygen species (ROS) produced by cellular changes, encourage the development of inflammatory factors [3]. PPARγ activation plays an important role and is crucial for controlling proliferation, differentiation, metabolism, development, and inflammatory reactions [4]. PPARγ has the ability to primarily prevent NF-κB activation [5,6] and also mediates the down-regulation of pro-inflammatory cytokines and interleukins IL-2, IL-6, TNF-α, and iNOS as depicted in Figure 1.

Numerous illnesses are influenced by inflammation [1], which is connected to the release of cytokines and oxygen-free radicals from activated inflammatory cells like eosinophils, neutrophils, macrophages, and monocytes. Proinflammatory chemokines and cytokines are upregulated during inflammation, and this activation and interaction with a variety of immunological factors result in inflammation [7,8]. Eugenol (4-allyl-2-methoxyphenol) is a phenolic substance belonging to the phenylpropanoid family. It is the principal constituent of clove (*Eugenia caryophyllata*) and belongs to the family Myrtacea [9]. The pharmacologically active compound eugenol is present in essential oils of plants [10]. It is also found in nutmeg (*Myristic fragrans*), basil (*Ocimum basilicum* L.), *Melissa officinalis* (Lemon Balm), cinnamon (*Cinnamomum zeylanicum*), *Ocium tenuiflorum* (Tulsi), and *Illicium anisatum* (Star Anise). Eugenol has proven to have many biological activities, including antibacterial [11], antioxidant [12,13], antitubercular [14], anticancer [15], antifungal [16], anti-inflammatory [17], and anesthetic activity [18]. The MEK phox pathway is inhibited by eugenol [19,20]. Several proinflammatory mediators, such as interleukins IL-5, interleukins IL-6, PGE_2_, Cox-2, and iNOS as well as activated NF-κB have all been reported to be inhibited by eugenol as shown in Figure 2.

Considering the importance of eugenol as a bioactive substance, we aimed to obtain novel derivatives of eugenol and to study possible anti-inflammatory activity. As a result, we synthesized eugenol-based PPARγ agonists analyzed, and screened for PPARγ protein binding, and then assessed their in vitro anti-inflammatory activity using an albumin denaturation assay and HRBC membrane stabilization method.

## 2. Results

### 2.1. Design of Eugenol Derivatives as PPARγ Agonists

The structural characteristics of PPARγ agonists include a heterocyclic head, typically thiazolidinedione, followed by a benzyloxy trunk, a two-carbon linker, and a lipophilic tail. Based on these structural characteristics, we designed novel eugenol derivatives with a benzyloxy trunk, two carbon linkers, and a lipophilic tail Figure 3. However, the allylic head is used in place of the heterocyclic head.

### 2.2. Synthesis

The rationally designed eugenol derivatives involved the formation of substituted 2-chloro-*N*-phenylacetamide **2** by treating various aromatic amines **1** with chloroacetyl chloride using dichloromethane as a solvent, following this the substituted 2-chloro-*N*-phenylacetamide **2** were then linked to eugenol in the presence of dry acetone and anhydrous potassium carbonate to produce novel eugenol derivatives (**1a**–**1f**) in good yields (70–87%) according to (Figure 1) [21]. By using the IR, ^1^HNMR, ^13^CNMR, and mass spectral techniques, the structures of synthesized compounds were confirmed and evaluated. The absence of the -OH group at 3350 cm^−1^ in IR spectra confirmed the formation of **2**. C=O and N-H groups were detected in the IR bands at 1690 cm^−1^ and 3460 cm^−1^, respectively. DMSO-d_6_ and CDCl_3_ were employed as the solvent for ^1^H-NMR and ^13^C-NMR. The chemical shifts are expressed as a number of parts per million (ppm), where s stands for singlet, d for doublet, t for triplet, and m for multiplet. All the synthesized compounds have a singlet in their ^1^H-NMR spectra that is attributed to the characteristic of the NH group and is present at about 8.73–9.97 ppm in addition to aromatic protons. one proton of CH= generates a multiplet at around δ 5.1–6.1 ppm and a doublet is produced by two protons of =CH2 at about δ 5.15 ppm. Similar to this, all the ^13^CNMR spectra show distinct signals attributed to the ketonic carbon at δ 167.6 ppm in addition to other typical carbon signals. ESI-MSMS was used to calculate the molecular mass of all the synthesized derivatives. All these facts support and confirm the formation of synthesized compounds (**1a**–**1f**)**.**

### 2.3. ADMET and TOPKAT Profile

For the purposes of determining the efficacies or dangers of small molecules, ADMET and toxic prediction of the compounds are key characteristics that must be evaluated [22]. The utilization of small molecular tools to predict the pharmacokinetics and dynamic profiles of synthesized compounds (Table 1). The ability of a drug to cross the blood–brain barrier and penetration is an important drug-related parameter. Nearly all compounds, including standard, have high blood–brain barrier penetration (level = 1), good human intestinal absorption (level = 0), low (level =2) to good (level = 3) solubility, and do not inhibit the activity of the CYP450 metabolizing enzyme. Furthermore, all the synthesized compounds obey Pfizer’s rule of 5 which is a key descriptor for oral bioavailability [23,24] ll compounds are free from mutagenicity and carcinogenicity as depicted in Table 1 and Table 2.

### 2.4. Structure-Based Drug Designing

The concept of structure-based drug designing was implemented for known compounds and drug-targeting proteins [25]. The ligand binding domain region of Peroxisome Proliferator-activated Receptor PPAR gamma amino acids such as Phe282, Cys285, Gln286, Arg288, His323, Tyr327, Leu330, Gly338, Met364, His449, and Leu469. Receptor–ligand interactions analysis shows that all compounds have the potential to bind at the ligand binding site in varying docking energy scores Table 3 [26]. The Cdocker energy is significantly high **1d**, **1f**, and Standard. The **1f** shows two hydrogen bonding interactions (Cys285, His323) with active site amino acid while **1d** interactions are shared with active site neighboring announced residues. Moreover, it is common for both compounds to show hydrophobic interactions with Cys285, Arg288, Met364, and His449. Similarly, **1e** binding energies are significant and high for **1f** compared with standard (Diclofenac). Further, both hydrophobic and hydrogen bond interactions are observed at the binding site of the egg lysozyme anti-inflammatory drug target Figure 4 and Figure 5.

### 2.5. Structure–Activity Relationships (SAR)

In order to synthesize multiple compounds for the better and more potent acquisition of molecules for various pharmacological targets, a structure–activity relationship (SAR) is an important tool and information source in organic chemistry [27]. The functional moieties connected to the compounds play a crucial role in predicting the compound’s biological activity from their chemical structure. In this study, interaction pharmacophores were employed to analyze the pharmacophore interactions in derivatives of receptor–ligand complexes (**1a**–**1f**). Interestingly, 4-Allyl-2-Methoxyphenoxy is common in all the compounds and shares two hydrophobic and one aromatic hydrophobic Figure 6. In contrast, in highly active compounds benzene ring shares ring aromatic, and the location of each functional group induces the biological activity. Benzene ring with chloride (**Cl**) substitution in ortho position (**1f**) is a highly active compound among all compounds in this library. In the meta position, the benzene ring’s chloride (**Cl**) substitution shows less biological activity. The halogen (**F**) replacement at the meta position of the bone ring shows a slight deviation in the activity, while the sole benzene and (**NO_2_**) functional group attached aromatic ring shows less activity. Hence the position of the ortho and meta-functional group is considered a crucial pharmacophore and directly proportional to the biological activity. The halogen group (**F**, **Cl**) interacts with active site residues of the drug targets and enhanced the anti-inflammatory effects in ppar gamma binding and egg lysozyme assay. Therefore, SAR relation is a powerful technique utilized in combinatorial chemistry to predict the functional and biological activities of the molecules [28].

In the Pharmacophore model: The orange color symbolizes the ring aromatic (RA), the green sphere is the hydrogen bond acceptor (HBA), the magenta sphere is the hydrogen bond donor (HBD), and the cyan sphere is the hydrophobic region.

### 2.6. Time-Dependent Parameter Conformational Analysis of the Complex

The binding modes of best-docked molecules of receptor-ligand were studied using a molecular dynamics simulation study for a simulation time of 1000ps. In order to assess the stability of the system, the geometric parameters of the protein–ligand complexes, such as the radius of gyration (Rg) and root mean square deviation (RMSD), were calculated [29]. The RMSD is used to measure the root mean square deviation of atomic positions of each conformation. The average distance between the atoms of various structural conformations of protein and ligand over a period of time. The average RMSD of the Cα atoms of Peroxisome Proliferator-activated Receptor (PPAR) gamma and heavy atoms of **1d** and **1f** was found to be 1.722 ± 0.07966 and 1.373 ± 0.03305 Å, respectively (Figure 7A). In contrast, the standard drug Pioglitazone with Peroxisome Proliferator-activated Receptor (PPAR) gamma complex had an average RMSD of 2.440 ± 0.07039 Å. Among the three complexes, **1f** complex deviations are lower than standard and **1d** which agrees with biological activity. Similarly, analyses of lysozyme bound complex average root mean square deviations are 1.518 ± 0.1071Å, 1.447 ± 0.07017 Å, and 2.268 ± 0.04858 Å (Figure 7B) of **1e**, **1f**, and standard drug (Diclofenac), respectively. Hence, a conformational change in the compounds has a direct influence on biological activity. 

Rg can be explained as the root mean square distance from each atom of the system to its center of mass [30]. The Rg values for protein-ligand complexes: Peroxisome Proliferator-activated Receptor (PPAR) gamma **1f** and Pioglitazone show fluctuations between 18.65 Å to 18.8 Å (Figure 7C). However, deviations are slightly less for **1d** ranging from 18.8 Å to 18.9 Å. Nevertheless, anti-inflammatory drug target lysozyme bound complexes Rg values are between 13.6 Å to 13.9 Å (Figure 7D) among **1f** complex 3D compactness shows least as compared with other complexes. The energy parameters analysis of all the complexes shows fewer variations in Table 4 that indicates energetically all three complexes are good Overall, the Peroxisome Proliferator-activated Receptor PPAR gamma-**1f** complex shows fewer deviations with time-dependent parameter analysis and **1f** Lysozyme complex shows best in inflammatory activity.

### 2.7. TR-FRET Assay

Using the Lanthasreen TR-FRET assay, the synthesized compounds were first preliminary tested for the PPARγ binding affinity [31]. A high 520/495 ratio is identified when the antibody is bound to the receptor because energy is transferred from the antibody to the fluorescent ligand (tracer) during this process [32]. A chemical under test displaces the tracer from PPAR-LBD, which lowers the TR-FRET ratio and decreases the FRET signal Figure 8 [33]. The compounds **1d** and **1f** showed a higher binding affinity for PPARγ among other synthesized compounds in comparison with standard pioglitazone. The assay results indicate that compounds **1d** and **1f** show an IC_50_ value of 6.47 µM and 5.15 µM in comparison with standard 1.77 µM, respectively. The IC_50_ values of synthesized compounds and standard drug pioglitazone are tabulated in Table 5.

### 2.8. In Vitro Anti-Inflammatory Activity

#### 2.8.1. Albumin Denaturation Assay

Inflammation is accompanied by protein denaturation. Tissue protein denaturation is a well-established cause of Inflammatory illnesses. Inflammatory disorders including rheumatoid arthritis, diabetes, and cancer have been linked to the denaturation of proteins. The ability of a substance to stop proteins from becoming denaturized has a role in the prevention of inflammatory disorders [34,35]. The possible anti-inflammatory activity of newly synthesized compounds and standard diclofenac was evaluated by the egg albumin denaturation assay method. Denaturation of proteins is accomplished by keeping the reaction mixture in a water bath at 70 °C for 10 min. As indicated in Table 6, demonstrated the effectiveness of synthesized compounds at different concentrations in preventing heat-induced albumin denaturation and maximum inhibition was observed for compound **1f** 50.221 ± 0.60 at 200µM when compared with standard diclofenac 49.469 ± 0.51 at 100 µM. Albumin denaturation was significantly inhibited by standard Diclofenac and synthesized compounds at their respective IC_50_ values (Table 7 and Figure 9).

When a protein is exposed to external stress such as a strong acid or basic, an organic solvent, or heat, albumin denaturation occurs, which results in the loss of the protein’s secondary and tertiary structure. Biological proteins lose their biological function when they are denatured. Therefore, a ligand’s capacity to prevent protein denaturation denotes the possibility of anti-inflammatory activity [36,37]. Albumin denaturation assay clearly showed anti-inflammatory properties. According to the current study, newly synthesized compounds can stabilize lysosomal membranes in vivo and inhibit the production of autoantigens caused by the denaturation of protein. From the results, it was observed that compounds **1d**, **1e**, and **1f** exhibit high activity when compared to other compounds. The active compound **1f** containing chloro group at the ortho position is the highly potent compound with an IC_50_ value of 67.02 µM when compared to **1e** containing chloro group and **1d** containing trifluoro group at the meta position with an IC_50_ values of 96.23 µM and 106.8 µM, respectively. The nitro group at para position **1b** is less active with an IC_50_ value of 294.3 µM than the methyl group at the ortho position **1c** with an IC_50_ value of 134.0 µM.

#### 2.8.2. HRBC Method

The principle of this procedure is to stabilize the membrane of human red blood cells by hypotonic membrane lysis. The release of lysosomal enzymes is a sign of inflammation. Lysosomal enzymes are released during the lysis of human red blood cells (HRBCs), and hence it is taken as a measure to evaluate the anti-inflammatory activity of synthesized compounds to reduce inflammation [38]. Lysosomal enzyme levels typically decrease as the HRBC membrane is stabilized. Therefore, the compounds that can cause HRBC membrane stabilization are expected to be anti-inflammatory [39]. In the current study, the activity of each synthesized compound was assessed as a percentage prevention of HRBC lysis in the concentration range of 6.25–400 µM. Diclofenac sodium was used as a standard drug. By analyzing the results, it can be concluded that all the synthesized compounds significantly reduced inflammation using the HRBC membrane stabilization approach. All compounds were found to have concentration-dependent membrane-stabilizing action. Compounds possessing chloro group at the ortho position **1f** and chloro group at the meta position **1e** as well as the trifluoro substitution in the meta position **1d** have generally stabilized the HRBC in the range of 67–85% as shown in Figure 10. Additionally, compounds **1a** with a single benzene ring and **1c** with a methyl substitution in the ortho position showed 60.88 and 62.54% prevention against HRBC lysis, respectively. The nitro substitution of compound **1b** has the least stabilizing effect. Compounds **1f** was found to exhibit the highest level of membrane stabilization with 85.79% protection against HRBC lysis in comparison to standard diclofenac sodium which showed 90.77% protection, respectively, at 400 µM Table 8. IC_50_ values of synthesized derivatives of eugenol and standard diclofenac are depicted in Table 9.

## 3. Discussion

We have developed a simple method to synthesize new eugenol derivatives (**1a**–**1f**) as PPARγ agonists based on the prediction of in silico results which we then analyzed with spectroscopic methods (FTIR, ^1^HNMR, ^13^CNMR, and Mass spectrometry). The novel derivatives obey the Pfizer rule and potentially all compounds are free from mutagenicity and carcinogenicity. Molecular docking was utilized to assess the designed compound’s ability to bind to the target protein. The docked structures at the binding sites were demonstrated by molecular dynamics simulations to be stable. All the newly synthesized compounds were found to be potential PPARγ agonists by molecular analysis and confirmed by in vitro TR-FRET protein binding assay. Thus, the synthesized compounds have proven to be a potential PPARγ agonist by molecular studies and in vitro assays. According to in vitro anti-inflammatory results, the synthesized compounds at concentrations ranging from 6.25 to 400 µM effectively reduced heat-induced albumin denaturation and exhibited maximum membrane stabilization with 85% protection against HRBC lysis in comparison to standard diclofenac sodium which showed 90% protection, respectively.

## 4. Materials and Methods

### 4.1. In Silico Studies and Spectral Data Analysis

The novel derivatives of eugenol were designed based on pharmacodynamics and pharmacokinetics requirements. BIOVIA, Discovery Studio 2019 (Dassault systems Biovia corp) was used to perform in silico prediction in order to comprehend the molecular behavior. Bayesian and regression models were also used to examine the compound’s established dosage range, mutagenicity, and carcinogenicity. All of the chemicals and solvents were purchased and procured from spectrochem (Lucknow, India) and Merck Ltd.(St. Louis, MI, USA) The melting points of the novel derivatives were established in open capillaries using a melting point instrument and are uncorrected. Aluminum plates that have been precoated with silica gel G are used to carry out TLC using n-hexane and ethyl acetate as the mobile phase (9:1) used to monitor the reactions and chemical purity and were made visible by UV rays. IR spectra are presented in cm^−1^ and were acquired using a shimadzu infrared FTIR spectrophotometer (FTIR-8400S, Shimadzu, Kyoto, Japan) utilizing KBr pellet technique. The Agilent VNMRS 400 equipment (Agilent 400 Hz, Agilent, Santa Clara, CA, USA) was used to record the ^1^H-NMR & ^13^C-NMR spectra with a chemical shift are given in ppm ranging from 0 to 10. Tetramethylsilane (TMS) is used as an internal standard. Coupling constants were recorded in hertz (Hz). Mass spectra were recorded using ESI-MSMS (Make-Waters USS, Model-Xevo G2-XS Q TOF, Make-Waters, Illinois, IL, USA). 

### 4.2. General Procedure for the Synthesis of Eugenol Derivatives

**Step-1:** Synthesis of substituted 2-chloro-*N*-phenylacetamide

Triethylamine (1.1 eq) and various aromatic amines (1 eq) were added to dichloromethane (80–100 mL) and transferred to a flask with a guard tube. A total of 1.1 eq of chloroacetyl chloride were added drop by drop over a period of 30 min and the mixture was agitated under chilled circumstances (0–5 °C). At room temperature, the reaction mixture was again agitated for 10–12 h. Thin-layer chromatography (TLC) was used to monitor the reaction with n-hexane and ethyl acetate as the mobile phase. After the completion of the reaction the mixture was treated with water and dilute HCl and transferred to a separating funnel and allowed to separate. The DCM layer was passed through anhydrous sodium sulphate after the water layer was removed. Substituted 2-chloro-*N*-phenylacetamide was obtained by evaporating the solvent.

**Step-2:** Coupling eugenol with substituted 2-chloro-*N*-phenylacetamide

Eugenol (1.2 eq), finely powdered anhydrous potassium carbonate (K_2_CO_3_ 3 eq), and 80 mL of dry acetone were added to a mixture of substituted 2-chloro-*N*-phenylacetamide (1 eq), and the mixture was stirred at 45 °C for about 26 h. As the reaction progressed, the spots were periodically checked using TLC with n-hexane and ethyl acetate as the mobile phase [40]. Ethyl acetate was used to extract the reaction mixture after the acetone was evaporated. After being washed three times with a 10% NaOH solution, once with water, and then once more with brine solution, the ethyl acetate layer was dried on anhydrous sodium sulphate as described in Figure 1. The final product was obtained by evaporating the ethyl acetate layer (Table 10).

### 4.3. The Spectral Data of Novel Synthesized Eugenol Derivatives (Appendix A)

#### 4.3.1. 2-(4-Alyl-2-methoxyphenoxy)-*N*-phenylacetamide (**1a**)

Brown liquid Yield: 80.0% Mol Formula: C_18_H_19_NO_3_. IR (KBr, cm^−1^): 3336.96 (-NH), 3072.71 (ArC-H), 2914.54 (AliC-H), 1689.70 (C=O), 1600.97 (ArC=C), and 1317.43 (C-O). ^1^H NMR (400 MHz, δ ppm, DMSO-d_6_): 9.977 (s, 1H, NH), 7.624 (d, 2H, ArH, *J* = 7.6 Hz), 7.319(t, 2H, ArH, *J* = 7.2 Hz), 7.072 (t, 1H, ArH, *J* = 7.6 Hz), 6.907 (d, 1H, ArH, *J* = 8.4 Hz), 6.826 (s, 1H, ArH), 6.701 (d, 1H, ArH, *J* = 8.0 Hz), 5.954 (m, 1H, =CH, *J* = 13.2 Hz), 5.077 (d, 2H, =CH_2_, *J* = 5.6 Hz), 4.995 (s, 2H, CH_2_), 3.776 (s, 3H, OCH_3_), and 3.348 (d, 2H, CH_2_). ^13^C NMR: (400 MHz, δ ppm, DMSO-d_6_): 166.92 (C=O), 149.32 (C-O), 145.86 (O-C), 138.46 (-CH=), 137.90 (C-NH), 133.97 (ArC), 128.87 (ArC), 123.75 (2ArC), 120.63 (2ArC), 119.58 (2ArC), 115.71 (=CH_2_), 112.89 (ArC), 69.04 (CH_2_-O), 55.69 (O-CH_3_), and 40.26 (CH_2_). ESI-MSMS (m/z) peak calculated for C_18_H_19_NO_3_ [M+H]^+^ 298.112 peak found [M+H]^+^ 298.102.

#### 4.3.2. 2-(4-Allyl-2-methoxyphenoxy)-*N*-(4-nitrophenyl)acetamide (**1b**)

Yellowish solid: Yield: 75.3% Mol Formula: C_18_H_18_N_2_O_5_. IR (KBr, cm^−1^): 3365.90 (-NH), 3076.56 (ArC-H), 2922.25 (AliC-H), 1612.54 (C=O), 1597.11 (ArC=C), 1508.38 (N-O), and 1336.71 (C-O). ^1^H-NMR (400 MHz, δ ppm, CDCl_3_): 9.384 (s, 1H, NH), 8.240 (d, 2H, ArH, *J* = 9.2 Hz), 7.793 (d, 2H, ArH, *J* = 9.2 Hz), 6.942 (s, 1H, ArH), 6.794 (d, 2H, ArH, *J* = 9.6 Hz), 5.972 (m, 1H, =CH, *J* = 11.2 Hz), 5.115 (d, 2H, =CH_2_, *J* = 6.4 Hz), 4.650 (s, 2H, CH_2_), 3.943 (s, 3H, O-CH_3_), and 3.363 (d, 2H, -CH_2,_
*J* = 6.8 Hz) ^13^C-NMR: (400 MHz, δ ppm, CDCl_3_): 172.823 (C=O), 154.834 (C-O), 150.621 (O-C), 148.850 (C-NH), 148.140 (C-NO), 142.040 (-CH=), 141.456 (2ArC), 130.160 (2ArC), 126.444 (ArC), 124.245 (2ArC), 121.238 (ArC), 117.882 (=CH_2_), 76.163 (CH_2_-O), 61.141 (O-CH_3_), and 44.903 (CH_2_). ESI-MSMS (*m*/*z*) peak calculated for C_18_H_18_N_2_O_5._ [M-H] 341.31 peak found [M-H] 341.30.

#### 4.3.3. 2-(4-Allyl-2-methoxyphenoxy)-*N*-(o-tolyl)acetamide (**1c**)

Colorless solid: Yield: 80.5% Mol Formula: C_19_H_21_NO_3_. IR (KBr, cm^−1^): 3388.77 (-NH), 3166.78 (ArC-H), 2934.47 (AliC-H), 1691.35 (C=O), 1597.17 (ArC=C), and 1377.05 (C-O). ^1^H-NMR (400 MHz, δ ppm, CDCl_3_): 8.739 (s, 1H, NH), 8.040 (s, 1H, ArH, 7.273 (d, 2H, ArH, *J* = 8.4 Hz), 7.120 (d, 1H, ArH, *J* = 7.2 Hz), 6.913 (d, 1H, ArH, *J* = 7.6 Hz), 6.786 (t, 2H, ArH, *J* = 7.6 Hz), 6.008 (m, 1H, =CH, *J* = 8.4 Hz), 5.135 (d, 2H, =CH_2_, *J* = 6.8 Hz), 4.668 (s, 2H, CH_2_), 3.890 (s, 3H, O-CH_3_), 3.385 (d, 2H, -CH_2_, *J* = 6.4 Hz), and 2.317 (s, 3H, -CH_3_). ^13^C-NMR: (400 MHz, δ ppm, CDCl_3_): 166.658 (C=O), 149.535 (C-O), 145.312 (O-C), 137.237 (-CH=), 135.233 (C-NH), 130.446 (ArC), 128.422 (2ArC), 126.827 (ArC), 125.037 (ArC), 122.098 (ArC), 120.795 (ArC), 115.959 (=CH_2_), 115.142 (ArC), 112.408 (ArC), 69.609 (CH_2_-O), 55.677 (O-CH_3_), 39.838 (CH_2_), and 29.690 (CH_3_). ESI-MSMS (*m*/*z*) peak calculated for C_19_H_21_NO_3_ [M+Na]^+^ 334.137 peak found [M+Na]^+^ 334.139. 

#### 4.3.4. 2-(4-Allyl-2-methoxyphenoxy)-*N*-[(3-trifluoromethyl)phenyl]acetamide (**1d**)

Dark brown liquid: Yield: 70.8% Mol Formula: C_19_H_18_ F_3_NO_3_. IR (KBr, cm^−1^): 3265.75 (-NH), 3083.43 (ArC-H), 2918.45 (AliC-H), 1687.30 (C=O), 1509.53 (ArC=C), and 1330.55 (C-O). ^1^H-NMR (400 MHz, δ ppm, CDCl_3_): 9.393 (s, 1H, NH), 7.990 (s, 1H, ArH) 7.887 (s, 1H, ArH) 7.459 (t, 1H, ArH, *J* = 7.6 Hz), 6.936 (d, 1H, ArH, *J* = 8.0 Hz), 6.865 (d, 1H, ArH, *J* = 8.2 Hz), 6.797 (d, 1H, ArH, *J* = 8.4 Hz), 6.707 (d, 1H, ArH, *J* = 9.6 Hz), 6.036 (m, 1H, =CH, *J* = 13.6 Hz), 5.163 (d, 2H, =CH_2_, *J* = 7.6 Hz), 4.657 (s, 2H, CH_2_), 3.933 (s, 3H, O-CH_3_), and 3.387 (d, 2H, -CH_2_). ^13^C-NMR: (400 MHz, δ ppm, CDCl_3_): 167.734 (C=O), 149.657 (C-O), 145.639 (O-C), 137.884 (-CH=), 135.958 (C-NH), 131.891 (ArC), 129.585 (ArC), 122.950 (ArC), 121.277 (ArC), 121.131 (ArC), 120.936 (ArC), 116.692 (=CH_2_), 116.568 (ArC), 116.033 (ArC), 114.554 (ArC), 112.666 (ArC), 70.733 (CH_2_-O), 55.828 (O-CH_3_), and 39.843 (CH_2_). ESI-MSMS (*m*/*z*) peak calculated for C_19_H_18_ F_3_NO_3_ [M+H]^+^ 366.101 peak found [M+H]^+^ 366.100.

#### 4.3.5. 2-(4-Allyl-2-methoxyphenoxy)-*N*-(3-chlorophenyl)acetamide (**1e**)

Brownish black liquid: Yield: 77.6% Mol Formula: C_18_H_18_ClNO_3_. IR (KBr, cm^−1^): 3342.75 (-NH), 3061.57 (ArC-H), 2985.62 (AliC-H), 1683.28 (C=O), 1593.86 (ArC=C), and 1307.78 (C-O). ^1^H-NMR (400 MHz, δ ppm, CDCl_3_): 9.143 (s, 1H, NH), 7.760 (d, 1H, ArH), 7.498 (d, 1H, ArH), 7.127 (t, 1H, ArH, *J* = 8.8 Hz), 6.932 (d, 1H, ArH, *J* = 8.0 Hz), 6.878 (d, 1H, ArH, *J* = 8.4 Hz), 6.812 (d, 1H, ArH, *J* = 6.8 Hz), 6.706 (d, 1H, ArH, *J* = 6.8 Hz), 6.029 (m, 1H, =CH, *J* = 10.4 Hz), 5.151 (d, 2H, =CH_2_, *J* = 8.4 Hz), 4.636 (s, 2H, CH_2_), 3.939 (s, 3H, O-CH_3_), and 3.385 (d, 2H, -CH_2,_
*J* = 6.4 Hz). ^13^C-NMR: (400 MHz, δ ppm, CDCl_3_): 167.432 (C=O), 149.676 (C-O), 146.582 (O-C), 137.865 (-CH=), 135.919 (C-NH), 131.891 (ArC), 130.257 (ArC), 126.278 (C-Cl), 124.555 (ArC), 121.257 (ArC), 119.963 (ArC), 116.821 (-CH=), 115.498 (ArC), 114.427 (ArC), 111.256 (ArC), 70.733 (CH_2_-O), 55.955 (O-CH_3_), and 39.872 (CH_2_). ESI-MSMS (m/z) peak calculated for C_18_H_18_ClNO_3_. [M+H]^+^ 332.087 peak found [M+H]^+^ 332.077.

#### 4.3.6. 2-(4-Allyl-2-methoxyphenoxy)-*N*-(2-chlorophenyl)acetamide (**1f**)

Colorless solid Yield: 87.9% Mol Formula: C_18_H_18_ClNO_3_. IR (KBr, cm^−1^): 3444.19 (-NH), 3062.52 (ArC-H), 2930.59 (AliC-H), 1676.47 (C=O), 1599.40 (ArC=C), and 1438.69 (C-O). ^1^H-NMR (400 MHz, δ ppm, CDCl_3_): 9.351 (s, 1H, NH), 8.456 (s, 1H, ArH), 7.401 (d, 1H, ArH, *J* = 7.2 Hz), 7.309 (d, 1H, ArH, *J* = 7.2 Hz), 7.085 (d, 1H, ArH, *J* = 7.2 Hz), 6.906 (d, 1H, ArH, *J* = 8.4 Hz), 6.769 (t, 2H, ArH, *J* = 6.0 Hz),), 6.009 (m, 1H, =CH, *J* = 6.8 Hz), 5.114 (d, 2H, =CH_2_, *J* = 6.0 Hz), 4.645 (s, 2H, CH_2_), 3.878 (s, 3H, O-CH_3_), and 3.364 (d, 2H, -CH_2,_
*J* = 6.4 Hz). ^13^C-NMR: (400 MHz, δ ppm, CDCl_3_): 167.014 (C=O), 149.812 (C-O), 145.279 (O-C), 137.252 (-CH=), 135.423 (C-NH), 134.139 (ArC), 129.157 (ArC), 127.649 (C-Cl), 124.935 (ArC), 123.300 (ArC), 121.588 (ArC), 120.635 (ArC), 115.935 (=CH_2_), 115.566 (ArC), 112.520 (ArC), 69.751 (CH_2_-O), 55.712 (O-CH_3_), and 39.843 (CH_2_). ESI-MSMS (*m*/*z*) peak calculated for C_18_H_18_ClNO_3_. [M+H]^+^ 332.121 peak found [M+H]^+^ 332.122.

### 4.4. ADMET, Drug Likeness and Toxicity Predictions

In order to assess in silico pharmacokinetics and pharmacodynamic studies, the synthesized compounds were submitted to ADMET and TOPKAT tools of the small molecule protocol [40]. Carcinogenicity, mutagenicity and the set dosage range of the synthesized compounds were analyzed using the Bayesian and regression models. Furthermore, Lipinski’s rule of five was applied to determine the compound’s oral bioavailability.

### 4.5. Molecular Docking

The macromolecule tool and small molecule protocol were used to prepare the structure of the drug-targeting protein and the compounds employed for docking [40]. The bound ligand coordinates 49.720X −36.98Y 19.294Z of radius 8.4 for PPARγ (2PRG) and 26.63X 5.50Y 14.22Z of radius 12 for egg lysozyme (3WXU) were defined as binding sites for docking with an equal grid spacing of 0.5 and 90-degree grid angles. The CDOCKER algorithm, a potent CHARMm-based docking method that has been shown to create accurate docked poses, was used to analyse receptor–ligand interaction. Every parameter was set to default while executing the docking process. Finally, the best pose of ligands was taken for molecular dynamics simulation and non-bonded interaction analysis. Consequently, the top-docked complexes were probed to estimate binding free energy using CHARMm implicit solvation models. The binding free energy GBSA is estimated between each ligand and the receptor.

### 4.6. Molecular Dynamics Simulation

Using Dassault System BIOVIA, Discovery Studio 2019 modeling Environment, 1000 Picosecond molecular dynamics simulations and the top two compounds of interaction with drug targets PPARγ and egg lysozyme were performed [40]. The process was done in five cascade steps, starting with two steps of 500-cycle energy minimization of a complex with the steepest descent and conjugate gradient. The system is then gradually heated from 50 to 300 K with 100 ps of simulation time and 500 ps of equilibration to reach degrees of freedom. The production using the canonical ensemble was then exposed to equal Tmass and Pmass at 300 K. In addition, the SHAKE constraint fixes all hydrogen-containing bonds in the simulation inside this cutoff range of 12Å–10 Å. Utilizing the leap-frog verlet algorithm, all atom velocities and positions are calculated at specific time points. Finally, RMSD and Rg were used to examine the deviations in complex conformations. 

### 4.7. TR-FRET Competitive Protein Binding Assay

Lanthascreen TR-FRET protein-binding competitive experiments were carried out using a buffer solution containing 50 mM TRIS, pH 7.5, 150 mM NaCl, 0.02% Triton^®^ X-100, and 5 mM DTT in a black flat bottom 384-microwell plates [40]. The ligand stocks and standard pioglitazone were made by serial dilution in 1% DMSO concentration, applied to wells in duplicates, and incubated for an hour at room temperature. A multimode low-volume proxy plate reader was used to read the plates after one hour of incubation. Then, a mixture of a human PPARγ- LBD labelled with glutathione S-transferase (GST), a fluorescent small pan-PPAR ligand (FluormoneTM Pan-PPAR Green), and terbium-labeled anti-GST antibody was added. The acceptor FITC emission was measured at 520 nm, the Tb source was stimulated at 340 nm and the Tb source’s fluorescence intensity was recorded at around 495 nm.

### 4.8. Anti-Inflammatory Activity

#### 4.8.1. Albumin Denaturation Assay

Minor modifications were made to Mizushima and Kobayashi’s and Sakat et al. methodologies [41]. The reaction mixture containing 0.2 mL of egg albumin (from a fresh hen’s egg) is mixed with 2.8 mL of phosphate-buffered saline (PBS, pH 6.4) and 2 mL of synthesized compounds in varying concentrations. The mixture was then heated at 70 °C for 5 min after being incubated at 37 °C for approximately 15–20 min. After cooling, their absorbance was measured at 660 nm spectrometrically [40]. Three duplicates of the experiment were carried out. The following formula was used to determine the percentage inhibition of protein denaturation.
Percentage inhibition = (OD control − OD sample/OD control) × 100

#### 4.8.2. HRBC Method

Healthy volunteer blood was drawn, and an equal volume of sterilized Alsevers solution (0.05% citric acid, 0.05% sodium citrate, 2% dextrose, and 0.42 % sodium chloride) was added. The packed cells in this blood solution were separated after centrifugation at 3000 rpm [41]. The packed cells were cleaned using an isosaline solution, and then, isosaline was used to prepare a 10% *v*/*v* suspension. This HRBC suspension was employed to estimate its anti-inflammatory properties. Different concentrations of the synthesized compounds and the reference standard were each combined with 1 mL of phosphate buffer (pH 7.4), 2 mL of hyposaline, and 0.5 mL of HRBC solution. A total of 2.0 mL of distilled water was utilized as the control. The assay mixtures were all centrifuged at 3000 rpm after 30 min of incubation at 37°C. The following formula was used to determine the percentage of stabilized HRBC membranes.
Percentage protection= 100 − (OD sample/OD control) × 100

### 4.9. Statistical Analysis

Statistical analysis was performed by one-way analysis of variance (ANOVA) using Graph pad prism 8.0 software. Data are expressed as the mean ± standard deviation of the mean. *p* values less than 0.05 were considered statistically significant.

## 5. Conclusions

Novel derivatives of eugenol were synthesized, analyzed, and screened for PPARγ protein binding assay and were evaluated for in vitro anti-inflammatory activity using an albumin denaturation assay and HRBC membrane stabilization method. The structural elucidation of synthesized compounds was accomplished by the use of IR, ^1^HNMR, ^13^CNMR and mass spectrometry. The designed compound’s capacity to attach to the target protein was evaluated using molecular docking. Molecular dynamics simulations showed that the docked structures at the binding sites were stable. From the TR-FRET protein binding assay, it was determined that all of the newly synthesized compounds were potential PPAR agonists. Our results show that the newly synthesized compounds have good anti-inflammatory properties by inhibiting the heat-induced albumin denaturation and red blood cells membrane stabilization at different concentrations. It was thought that compound **1f** was a bioactive substance, further studied at a molecular level, and can be investigated to learn more about its pharmacological action.

## Data Availability

Not applicable.

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
