# Peer review of "Novel Derivatives of Eugenol as a New Class of PPARγ Agonists in Treating Inflammation: Design, Synthesis, SAR Analysis and In Vitro Anti-Inflammatory Activity"

_molecules, 2023, doi:10.3390/molecules28093899_

Round 1

Reviewer 1 Report

The set of compounds 1a-1f, were synthesized from eugenol and characterized using FT-IR, 1H NMR, 13C NMR, and MS techniques. The compound 1f shows high anti-inflammatory activity and inhibit PPARγ. 

This work should be improved according to the following comments:

1) L. 97: correct "Cdcl3" also in the spectral data: "1H-NMR (400MHz, δ ppm, Cdcl3)"

2) Check the tables, why do the values have a different number of decimals?

3) Check the codes of compounds, they should be in bold.

4) L. 243: Why do you compare compound 1f and diclofenac at different concentration?

5) Section 4.1. In-silico studies and spectral data analysis. Models must be specified for all instruments and devices.

6) Table 8: You should specify the solvent system for all Rf values.

7) L. 478: Correct the percentage inhibition formula.

Author Response

We thank the editor and reviewers for the comments and suggestions to improve the scientific quality of the manuscript. We have addressed all the comments and incorporated changes in the revised manuscript.

The modifications in the revised manuscript are marked using the track change’s function.

Reviewer 2 Report

1- The authors have to represent the function of each protein used in the docking as a receptor.

2- Authors should do one of the validation methods of the docking process

Author Response

(The authors gave the same response as above.)
